# Online pragmatic interpretations of scalar adjectives are affected by perceived speaker reliability

Bethany Gardner[1☯], Sadie Dix[2☯], Rebecca Lawrence[3], Cameron Morgan[3], Anaclare Sullivan[4], Chigusa Kurumada[3]*

1 Department of Psychology and Human Development, Vanderbilt University, Nashville, Tennessee, United States of America, 2 Friedman School of Nutrition Science and Policy, Tufts University, Boston, Massachusetts, United States of America, 3 Department of Brain and Cognitive Sciences, University of Rochester, Rochester, New York, United States of America, 4 Department of Epidemiology and Biostatistics, University at Albany, Albany, New York, United States of America

☯ These authors contributed equally to this work.
* ckuruma2@ur.rochester.edu

**Data Availability Statement:** The data file, analysis scripts, and visual stimuli are available from the Open Science Framework repository: 10.17605/OSF.IO/QTY25.

## Abstract

Linguistic communication requires understanding of words in relation to their context. Among various aspects of context, one that has received relatively little attention until recently is the speakers themselves. We asked whether comprehenders' online language comprehension is affected by the perceived reliability with which a speaker formulates pragmatically well-formed utterances. In two eye-tracking experiments, we conceptually replicated and extended a seminal work by Grodner and Sedivy (2011). A between-participant manipulation was used to control reliability with which a speaker follows implicit pragmatic conventions (e.g., using a scalar adjective in accordance with contextual contrast). Experiment 1 replicated Grodner and Sedivy's finding that contrastive inference in response to scalar adjectives was suspended when both the spoken input and the instructions provided evidence of the speaker's (un)reliability: For speech from the reliable speaker, comprehenders exhibited the early fixations attributable to a contextually-situated, contrastive interpretation of a scalar adjective. In contrast, for speech from the unreliable speaker, comprehenders did not exhibit such early fixations. Experiment 2 provided novel evidence of the reliability effect in the absence of explicit instructions. In both experiments, the effects emerged in the earliest expected time window given the stimuli sentence structure. The results suggest that real-time interpretations of spoken language are optimized in the context of a speaker identity, characteristics of which are extrapolated across utterances.

## Introduction

One of the essential properties of linguistic communication is its capacity to convey meanings beyond what is explicitly stated [1]. Speakers cannot, and do not, encode every detail of the information they are intending to communicate. Comprehenders therefore need to integrate

**Funding:** We would like to declare that the authors received funding from the Department of Brain and Cognitive Sciences at the University of Rochester. The funder had no role in study design, data collection and analysis, decision to publish, or preparation of the manuscript.

**Competing interests:** The authors have declared that no competing interests exist.

extra-sentential sources of information (e.g., local context, common ground, world knowledge) to infer the speaker's pragmatic intentions [2–5]. There is now a great deal of evidence that the preceding linguistic discourse [6–9], visual context ([10–12] but see [13]), and task constraints [14, 15] can shape pragmatic inference during online language understanding (see [16, 17] for reviews). Many questions remain open, however, about what constitute a context and when a particular context is considered in language processing.

The goal of the current study was twofold. First, we conducted a conceptual replication of a seminal study in this domain conducted by Grodner and Sedivy [18]. They proposed that listeners consider a speaker, and the reliability with which they produce pragmatically well-formed utterances, as part of contextual information. As we describe below, the importance of Grodner and Sedivy's work has now been firmly established. The experimental results, however, have not been widely replicated (but see [19]). The second goal was to extend the insight of Grodner and Sedivy [18] to ask a novel question: Is a top-down, explicit instruction to declare the pragmatic (un)reliability of a speaker necessary to trigger a modulation of pragmatic processing? This relates to a broader scope of inquiries into the nature and effects of a speaker as context. Do listeners constantly assess and integrate speaker information in their processing? Or alternatively, do they consider it only under special circumstances where the need to do so is explicitly asserted?

Grodner and Sedivy's [18] experiment built on an earlier, now classic, study that illustrated the significance of context and pragmatic knowledge in language understanding. In a visual-world eye tracking experiment, Sedivy and her colleagues [20] asked participants to manipulate objects based on spoken instructions such as "Pick up the tall glass". An array of objects consisted of a target item (e.g., a tall glass), a competitor (e.g., a tall pitcher), a contrast (e.g., a short glass), and a distractor. The results showed that the partial instruction "Pick up the tall —" elicited an increase in fixations to the tall member of the contrast pair (e.g., the tall vs. short glasses) rather than the other tall object (e.g., the pitcher) in the display. This is taken as evidence that listeners are interpreting the scalar adjective (e.g., tall) in relation to the size contrast present in the given context. Indeed, no such early fixations were observed when the display did not contain a contrast item (e.g., only the tall glass and the tall pitcher present, no short glass). This seemingly simple example illuminates the flexibility and speed with which comprehenders integrate contextual information into the resolution of referential ambiguity. It also establishes anticipatory looks to a target as an important tool that provides a window into the real-time pragmatic interpretation of scalar adjectives (see also [21–23]).

One remaining question of theoretical significance was whether the early fixations, as observed in [20], indeed reflected comprehenders' pragmatic (as opposed to semantic) interpretation of a scalar adjective, generated in a given context, on the fly [24]. This is the question Grodner and Sedivy [18] set out to address. If the early fixations are pragmatic in nature, such inference should be defeasible i.e., canceled without contradiction when a context does not support it. If, on the other hand, the fixations are attributable to the comprehender's semantic knowledge of scalar adjectives, they should persist across contexts. As a means to probe the defeasibility, they manipulated the pragmatic cooperativeness ("reliability") of the speaker. Using a between-subject design, they introduced a "reliable" speaker, who conformed to the general cooperative principle of communication [1], providing necessary and sufficient modification to pick out a target referent in a scene. To a different group of comprehenders, they introduced an "unreliable" speaker, who "suffers from a communicative and language impairment" and thus was unlikely to formulate referential expressions with contextual demands in mind. They found that comprehenders exposed to the reliable speaker generated the early fixations in response to a scalar adjective, as per [20], while those exposed to the unreliable speaker did not.

This and similar studies have demonstrated that a speaker's communicative abilities and their knowledge states constitute an important contextual backdrop for language processing. Arnold, Hudson-Kam, and Tanenhaus [25], for instance, found that comprehenders stopped interpreting an instance of a disfluency (such as "uh" or "um") as a cue to an infrequent object label when the speaker was believed to suffer from object agnosia (see also [26]). Expectations for the speaker's reliability, cooperativeness, and expertise have been found to modify a pragmatic interpretation that would otherwise be evoked [22, 25, 27–31]. Importantly, as in [18, 25], some of these effects were detected in *real-time* (online) language processing as well as in offline judgments. In some hegemonic views on pragmatic interpretation of speech, all contextual information is registered and processed after a semantic meaning of an utterance is derived ([19, 31–33], for a review, see [34, 35]). The rapid and immediate impact of speaker information on the interpretation of a scalar adjective runs counter to this view. It instead suggests that at least some contextual information can be integrated synchronously into processing as the input unfolds over time [36].

To further expand our understanding of the significance of speaker-information in pragmatic processing, we need to address two limitations of the extant body of work. One limitation concerns replicability of study results. As we discuss below, the influential work by Grodner and Sedivy [18] included design and analysis features that might limit the generalizability of their results. Importantly, the original study involved a human experimenter providing live instructions. Although the interactive nature of the experiment was important in examining pragmatic inference over characteristics of the speaker, it reduced the amount of control over stimuli, resulting in increased variability across trials and subjects. Here we provide a conceptual replication of their experiment using pre-recorded speech with some modifications in technical details of experimental and analysis procedures.

Second, it is not as yet clear whether, and if so to what extent, comprehenders can spontaneously integrate speaker-information without an explicit instruction to do so. Most of previous studies on the topic of speaker information used explicit instructions (e.g., "This speaker suffers from an impairment") to establish the pragmatic trait relevant to their manipulation. Such explicit characterizations of the speaker are, however, rare in our ordinary linguistic communication. More common are situations in which comprehenders have a perceived sense of the speaker's pragmatic (un)reliability from the observed productions. If the speaker-reliability effect is observed only in the presence of an explicit instruction, its importance and validity in naturalistic language uses will be limited. In the current study, as we describe below, we construct a new condition in which comprehenders encounter the linguistic evidence of speaker (un)reliability while receiving no explicit prompts to anticipate any anomaly in referential expressions.

## Current study

The goals of the current study were thus twofold. First, we aimed to conceptually replicate the experiment of Grodner and Sedivy [18]. Despite its considerable impact, their reliability manipulation has been replicated by only a few [19, 37]. The present experiments were also designed to complement the strengths and weaknesses of the original study:

1. Whereas the original study used confederate human interlocutors, we employed a computer-based paradigm. While human interlocutors can increase the ecological validity of an experiment, there are also procedural challenges stemming from more spontaneous interactions (for a review, see [38]). Of particular relevance for the present purpose, our computer-based paradigm ensures that the delivery and timing of the input are *identical* across participants, differing only in the intended between-participant manipulations regarding speaker reliability.

2. The original study contained an additional (unrelated) manipulation to test effects of deictic expressions (D. Grodner, personal communication), which we excluded. This approach in the present paper reduced the number of statistical tests to be conducted and provided a direct test of the reliability effect on the interpretation of scalar adjectives.

3. The original study, as well as that of Sedivy et al. [20], used a visual display with one or no contrast item present (Fig 1). This method rendered the no-contrast trials "over modified" for both the pragmatically reliable and unreliable speakers. As illustrated in the left panel in Fig 1, the adjective "large (cup)" was used in a context where there is only one cup present. This means that even in the reliable speaker condition, comprehenders were repeatedly exposed to over-informative instructions. Although this itself does not confound the original conclusion, it can neutralize differences between the reliable and unreliable speaker conditions. To more cleanly separate these conditions, the present design replaced no-contrast visual displays with 2-contrast displays (as used in [15, 17]). The 2-contrast scenes do not allow the comprehender to identify the intended referent on the basis of the adjective alone (e.g., "large" could be compatible with the large cup and large apple), thus serving as a baseline to measure the contrast effect in a 1-contrast visual display.

4. The eye-movement analysis of the original paper focused on the time window of 500ms beginning 200ms after the adjective *offset*, an approach which is not well motivated for assessment of anticipatory eye movements. (We discuss this issue in the general discussion.) We instead used a 500ms window beginning 200ms after the adjective *onset* to be able to detect the earliest effects of contrastive interpretations of an adjective. This analysis window was chosen based on a conservative, empirically attested estimate of the earliest linguistically mediated saccades in the same type of four-picture display [39, 40].

We also aimed to assess whether explicit instructions about the pragmatic traits of the speaker were required to cause speaker-dependent modulation of contrastive inference. In other domains of language processing such as speech perception, it has been demonstrated

**Fig 1. Example displays (no-contrast, 1-contrast vs. 2-contrast conditions) for the instruction, "Click on the large cup".** The no-contrast condition was used in Grodner and Sedivy [18], but replaced with 2-contrast condition in the current experiment. This figure is original, compliant with the CC BY4.0 license, produced by the current study team based on the information provided by Grodner and Sedivy [18]. The images for the 1-contrast and 2-contrast visual displays are also original. They are similar, but not identical, to those that were used in the eye-tracking studies. They are presented here for illustrative purposes only.

that comprehenders are sensitive to speaker-specific differences even in the absence of explicit instructions (e.g., [41–43], for reviews see [44, 45]). Evidence from speech perception also suggests that speaker-dependent perception can emerge rapidly, after only a couple of minutes of exposure [46, 47]. In the domain of real-time pragmatic processing, it remains an open question whether comprehenders can modulate their inferences based on the information in the linguistic input alone. Answering this question provides us with a window into the process with which comprehenders adjust their moment-by-moment interpretations of the linguistic input in context.

Finally, we conducted post-hoc analyses that assess how quickly the effects we observe emerged across trials. Although this was not an original goal of our study, addressing this question can shed light on underlying mechanisms responsible for adjustments of real-time language comprehension to pragmatic reliability and other characteristics of talkers. The overall patterns of results present intriguing questions to be addressed in future work.

## Experiment 1

The aim of Experiment 1 was to extend Grodner and Sedivy [18] to test whether comprehenders modulate their interpretation of scalar adjectives based on the speaker's pragmatic reliability.

### Method

**Participants.** A total of 48 undergraduate students from the University of Rochester participated in the current study. All participants were native speakers of American English with normal or corrected-to-normal vision and normal hearing. The experiment lasted roughly 30 minutes and participants received $10. The study protocol was approved by the University of Rochester Research Subjects Review Board (Protocol number: 00057827). Written consent was obtained from all participating subjects.

**Materials.** Experiment 1 consisted of 52 trials (4 example, 16 critical, and 32 filler trials) in which participants saw a 2 x 2 grid of pictures of animals and common objects in conjunction with an auditory instruction regarding which item to click on (Fig 1) (Experimental stimuli, together with data and analysis scripts, are available through DOI 10.17605/OSF.IO/QTY25). The visual stimuli were vetted through a prior experiment with 72 participants [48]. The instructions had the structure "Click on the X." All participants saw eight critical trials in the 1-contrast condition, in which the instruction contained the scalar adjective "large" or "small" (e.g., "Click on the large cup"). The display contained a set of size-contrasting items, one of which was the target item (e.g., a large and a small cup), and two distractors (e.g., a large apple and a small scarf). Four items included the adjective *large*, and the other four included *small*. Another eight critical trials comprised the 2-contrast condition, in which the auditory stimulus again contained a scalar adjective (e.g., "Click on the large cup"), but the display contained two sets of size-contrasting items, one pair that contained the target (e.g., a large and a small cup) and one pair that did not (e.g., a large and a small apple). As in the 1-contrast condition, half of the critical items included the adjective *large*, and the other half included *small*. Assignments of a given target item (e.g., a large cup) to the contrast conditions (i.e., 1- vs. 2-contrast) were counterbalanced across participants. That is, no participant saw a given item in both 1- and 2-contrast conditions.

Visual stimuli in the 32 filler trials were identical across all the participants. 50% (16 trials) of the filler trials belonged to the 1-contrast condition, and the rest belonged to the 2-contrast condition. In the 1-contrast filler trials, target items were one of the singletons to mask the contrast manipulation (e.g., *apple* or *scarf* in the example in Fig 1). Spoken instructions for the filler trials were manipulated between the reliable- and unreliable-speaker conditions: Those

in the reliable-speaker condition were optimally informative in encoding the size information as necessary (i.e., no adjective to refer to a singleton). Those in the unreliable-speaker condition had the following features: 75% (24/32) contained an additional, superfluous modifier (e.g., "Click on the small, pretty doll," when only one doll was present). 12.5% (4/32) were under-informative instructions (e.g., "Click on the doll," when two dolls were present), and 12.5% (4/32) contained an incorrect noun (e.g., "Click on the large toothbrush," with a hairbrush on the screen).

Inclusions of these multiple types of pragmatically "non-optimal" instructions were motivated by the original manipulation by Grodner and Sedivy [18], which included: a) explicit instructions characterizing the speaker as suffering from an "impairment that caused language and social problems"; b) a small number (4% of the totaling 200 instructions) of erroneous labels and locations; and c) a large number of over-informative adjective use (88% of cases where a bare-noun phrase would have been sufficient for a unique reference). Using these distinct types of instructions makes it difficult to pinpoint which of these make the speaker look most unreliable. We nonetheless decided to include all the kinds in order to conceptually replicate the original study, as well as to maximize the chance of participants picking up on the pragmatic anomaly.

Four counterbalancing lists for each speaker condition were created by crossing: 1) a given target item appearing in either the 1- or 2-contrast condition, 2) order of trials (forward vs. backward).

**Procedure.** Participants were randomly assigned either to the reliable- or unreliable-speaker condition. They first listened to one of the two types of explicit instructions from Grodner and Sedivy's [18] study, introducing the task and the speaker (S1 Appendix). Importantly, in the unreliable-speaker condition, participants learned that "(T)he study is intended to examine communicative aspects of his language impairment," while no such mention was made in the reliable-speaker condition. As part of the introduction, participants watched a video clip in which the speaker produces instructions for the four example trials. The speaker was introduced as a participant in a related communication study, referring to a target item in a four-picture display. In the reliable-speaker condition, the speaker used concise referring expressions, such as an unmodified expression for a singleton (e.g., "Click on the ladybug") and a modified expression for an item in a contrast set (e.g. "Click on the small banana"). In the unreliable-speaker condition, the speaker used redundant modifiers for a singleton (e.g., "Click on the small, spotted ladybug") and a double-modified expression for an item in a contrast set (e.g. "Click on the small, yellow banana"). We employed these video examples to provide an embodied image of the speaker, an approach which is expected to scaffold participants' pragmatic inferences.

Each of the 48 trials following the examples began with the presentation of the display. After one second of visual preview, participants heard a spoken sentence over Sennheiser HD570 headphones and clicked on the referent that best matched the sentence. Eye movements were monitored using a head-mounted SR Research EyeLink II system sampling at 250 Hz, with drift correction procedures performed every fourth trial.

## Results

Participants selected the correct target item in almost all trials. In the 768 critical trials measured (48 participants * 16 critical trials), only one was an error in which a non-target item was selected. This trial was removed from the subsequent analyses.

**Effects of reliability and contrast.** We examined participants' interpretation of the scalar adjective, indexed by the proportion of fixations they made to the target item in response to a

scalar adjective in the instruction (e.g., "Click on the <u>large</u> cup"). Plots of the full time-course of fixations by condition are provided in Fig 2. We first down-sampled the eye-tracking samples into 48ms bins (12 samples per bin at 250 Hz sampling). For each time bin, we calculated the proportion of fixations to the target, competitor, and contrast separately for each unique combination of participant and item (768 = 48*16 combinations per bin). Finally, we averaged these proportions for each time bin and then for each participant. As in other psycholinguistic experiments of this type, the current data included a larger amount of cross-participant, compared to cross-item, variability in the dependent variable. Using the by-participant averages as the input to the plots ensures that the confidence intervals shown in Fig 2 are not anti-conservative.

We then calculated target fixations in the planned analysis time window (i.e., a 500ms window starting 200ms after the adjective onset). As can be seen in Fig 2B, this analysis window covered on average 256ms of adjective duration and 244ms of noun duration. Given that it takes approximately 200ms to plan and launch eye-movements in response to linguistic input, data within this analysis window can be safely assumed to reflect comprehenders' response to the adjective. We first calculated the average proportion of fixations to the target for each unique combination of participant and item. For analysis, these trial-level proportions of target fixations were first transformed into empirical logits [49] and then analyzed in a linear mixed model using the *lme4* software package in R [50], R version 3.5.0 (2018-04-23). Each case was weighted by the inverse of its expected variances (following Barr, 2008). We included contrast condition (sum-coded: 1 = 1-contrast vs. -1 = 2-contrast) and reliability (sum-coded: 1 = reliable vs. -1 = unreliable) along with their interaction as fixed effects. The analysis converged

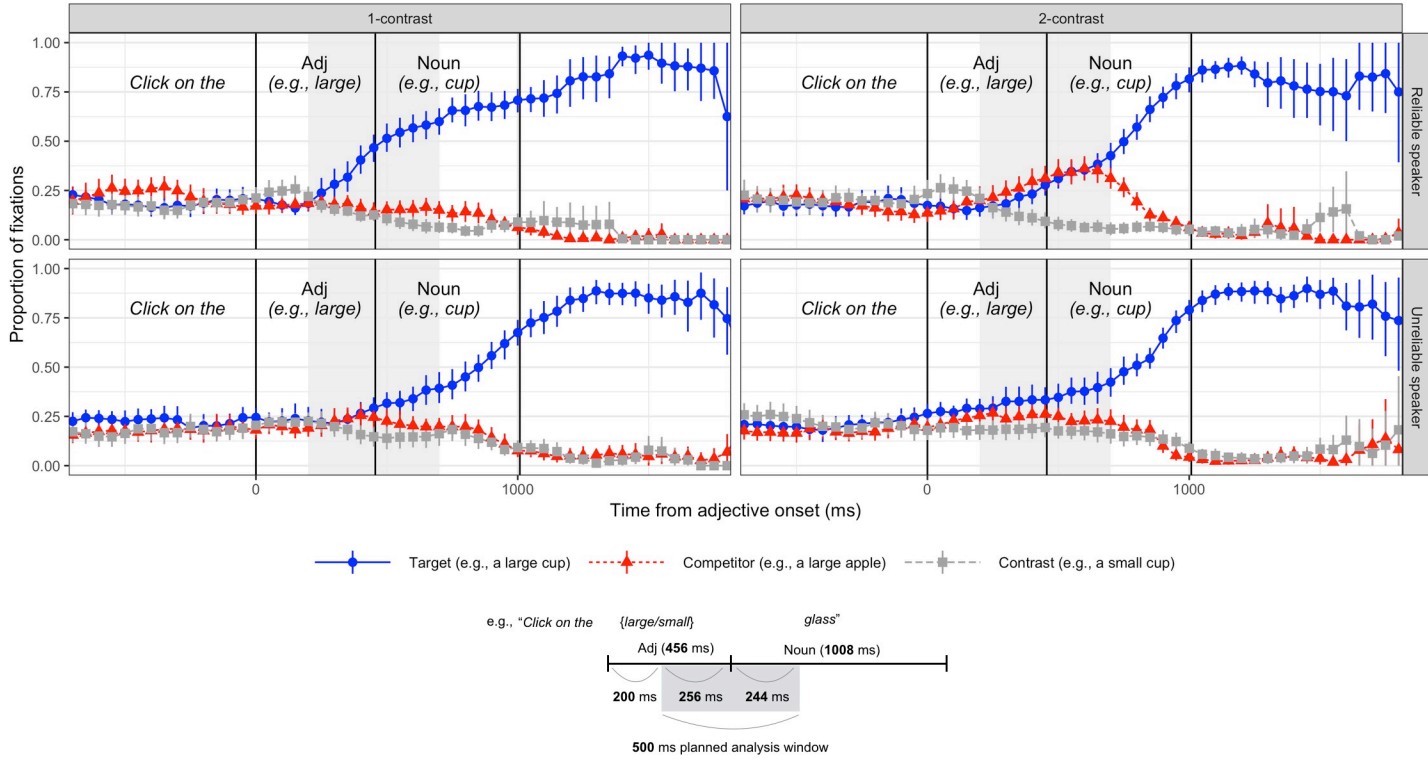

**Fig 2. A.** Time-course of fixations during instructions relative to the onset of the adjective (e.g., Click on the <u>large</u> cup) in the reliable (top) and unreliable (bottom) speaker conditions in Experiment 1. Vertical lines indicate the average adjective onset, adjective offset, and noun offset. Shaded area indicates the 500 ms planned analysis window. Points represent averages of by-participant means in each time bin. Error bars represent 95% bootstrapped confidence intervals over those by-participant means. **B.** Average duration of the scalar adjectives and the final nouns and their alignment with the 500ms planned analysis window.

with the maximum random effect structure justified by the design: by-participant and by-item random intercepts, by-participant and by-item random slopes for contrast, and by-items random slopes for reliability and the contrast x reliability interaction. A summary of the model output is provided in Table 1. *P*-values were obtained using a *t*-test with Kenward-Roger's approximated degrees of freedom, as implemented in *lmerTest* [51]. We calculated the by-participant averages for each combination of the speaker and contrast conditions (averaging across items within each participant).

No significant main effects of reliability or contrast on target fixations were observed. Crucially, the interaction term between reliability and contrast conditions was significant ($\beta$ = .065, $t$ = 4.412, $p$ < 0.001), suggesting that participants were more likely to make anticipatory eye movements based on a scalar adjective in 1-contrast trials than in 2-contrast conditions, particularly when exposed to the reliable-speaker. A simple effect analysis revealed that the effect of contrast was significant in the expected direction for participants in the reliable-speaker condition ($\beta$ = .091, $t$ = 4.372, $p$ < 0.001), replicating the original finding reported by Sedivy et al. [20]. For participants in the unreliable-speaker condition, the effect of contrast was not significant.

## Discussion

The reliability manipulation yielded a significant difference between the two reliability conditions in the amount of target fixations based on a scalar adjective. We thus successfully extended Grodner and Sedivy's original findings, while using the modified contrast conditions (i.e., 1- vs. 2-contrast visual displays) on a computer-based paradigm. With a pragmatically unreliable speaker, participants did not immediately fixate the target item (e.g., large glass) even when the adjective could uniquely single out the referent in a context. This supports that online pragmatic inferences can be modulated according to the likelihood that a speaker makes a contextually supported, contrastive use of prenominal adjectives in reference.

The reduced effect of contextual contrast in the unreliable-speaker condition could stem from two sources: The explicit instruction about the speaker's pragmatic impairment and/or the bottom-up input highlighting the unconventional traits of the speaker's language use. We conducted Experiment 2 to address the question of whether the instruction about the speaker was necessary to trigger the modulation of eye movements observed in Experiment 1. In other words, Experiment 2 examined whether there is a mechanism with which comprehenders spontaneously derive assessments of the reliability of a given speaker and modulate their anticipatory eye movements in subsequent instances of language comprehension.

## Experiment 2

The goal of Experiment 2 was to test whether the bottom-up input alone would be sufficient to trigger modulation of online pragmatic interpretations of scalar adjectives. We present joint

**Table 1. Summary of final linear mixed model of fixation proportions in Experiment 1.**

| Experiment 1: Fixed effects: | | | | |
|---|---|---|---|---|
| | $\beta$ | Std. Error | *t*- value | *p*-value |
| (Intercept) | -2.50 | .028 | -90.476 | < 2e-16 |
| Reliability (reliable vs. unreliable) | .015 | .021 | .732 | n.s. |
| Contrast (1-contrast vs. 2-contrast) | .026 | .014 | 1.75 | n.s. |
| Reliability X contrast | **.065** | **.015** | **4.412** | **< .0001** |

Number of observations: 768; Participants: 48; Items: 16.

analyses over the data from Experiments 1 and 2, including trial-level analyses that aim to assess how quickly the effect of speaker reliability emerged during the experiment.

## Methods

**Participants.** A new group of 24 undergraduates from the University of Rochester participated in Experiment 2. The criteria and compensation for participants were identical to Experiment 1.

**Materials.** All materials for the example, critical, and filler trials were identical to those in Experiment 1.

**Procedure.** There was only one condition in Experiment 2. All 24 participants were presented with the verbal instruction and introduction video identical to those in the reliable-speaker condition in Experiment 1. They were then exposed to four example items from the unreliable-speaker condition and the 48 experimental materials from the unreliable speaker condition in Experiment 1 (Table 2).

## Results

Participants selected the correct target item in all 384 critical trials measured (24 participants * 16 critical trials).

**Effects of reliability and contrast.** The analysis approach was identical to Experiment 1. Plots of the complete time course of target, competitor, and contrast fixations by condition are provided in Fig 3. As in Experiment 1, we computed target fixations in the predetermined time window (i.e., the 500 ms window starting 200 ms after the adjective onset). The average proportions of target fixations within this time window across reliability and contrast conditions are shown in Fig 4.

To evaluate the effect of the reliability manipulation, we combined the data from Experiment 1 and Experiment 2, resulting in three types of reliability conditions: reliable, unreliable with explicit instruction (Experiment 1), and unreliable bottom-up only (Experiment 2). As shown in Table 3, we used Helmert coding to compare 1) the unreliable conditions against the reliable condition (i.e., $C_1$); and 2) the two unreliable conditions against each other (i.e., with bottom-up information only vs. with explicit instructions, $C_2$).

As in Experiment 1, empirical logit-transformed proportions of target fixations were analyzed in a linear mixed model with contrast (sum-coded as in Experiment 1), reliability, and their interactions as fixed effects. The model converged with the maximum random effect structure justified by the design (see Experiment 1). A summary of the model output is provided in Table 4.

The main effects of reliability and contrast were not significant. Importantly, the interaction between contrast and the reliable-speaker vs. the two unreliable-speaker conditions was

**Table 2. Combinations of instructions and bottom-up manipulations in Experiments 1 and 2.**

| Reliability condition | Explicit instruction | Bottom-up information (filler items) |
|---|---|---|
| Reliable | reliable (no mention of the communicative impairment) | reliable (concise and contextually calibrated uses of modifiers) |
| Unreliable (Exp 1) | unreliable (mention of the communicative impairment) | unreliable (including over- and under-modifications and mislabels) |
| Unreliable (Exp 2) | reliable (no mention of the communicative impairment) | unreliable |

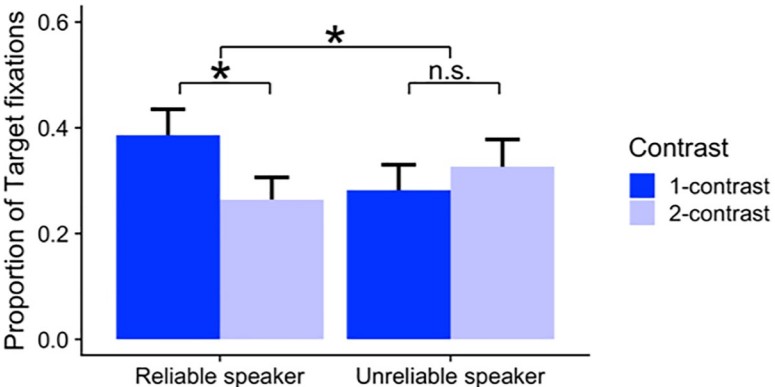

**Fig 3. Time course of target fixations during instructions (e.g., Click on the large cup) in the unreliable-speaker condition with only bottom-up information (Experiment 2).** The 1- and 2-contrast conditions are plotted in the left and right panels, respectively. Vertical lines indicate the mean 1) adjective onset (aligned to 0), 2) adjective offset, and 3) noun offset. Shaded area indicates the 500ms planned analysis window.

significant ($\beta$ = .078, $t$ = 4.623, $p$ < .0001): participants in the reliable-speaker condition—compared to those in the two unreliable-speaker conditions—were more likely to generate anticipatory eye movements in response to the scalar adjectives used in the 1-contrast condition. On the other hand, the interaction between contrast and the unreliable with bottom-up information only vs. unreliable with explicit instructions was not significant. This result suggests that the two unreliable-speaker conditions did not differ significantly from each other. A follow-up simple effect analysis revealed that the effect of contrast was *not* significant either in the unreliable speaker condition with only bottom-up information (Experiment 2) or in the unreliable-speaker condition in Experiment 1. In both of these conditions, the effect of contrast exhibited a numerical trend in the direction opposite from the reliable-speaker condition. We expand upon this point in the general discussion.

## Discussion

Upon repeated exposure to over- and under-informative instructions, participants exhibited behavioral responses similar to those which have been observed in the presence of explicit, salient instructions (e.g., "This speaker suffers from communicative and linguistic

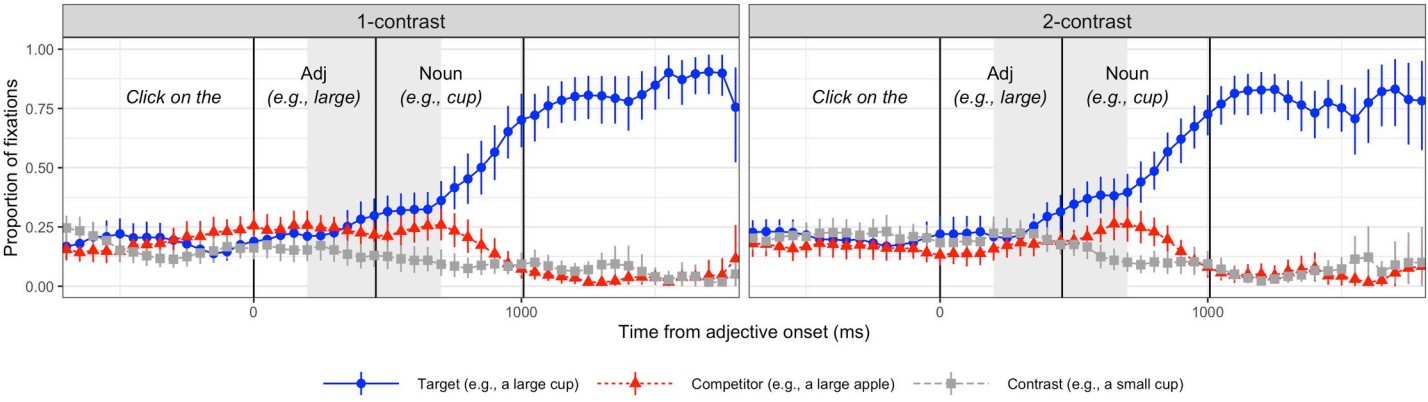

**Fig 4. Average proportions of target fixations during the 500ms window starting 200ms after the adjective onset (e.g., *Click on the large cup*) by reliability and contrast conditions in Experiments 1 and 2.** Error bars represent 95% bootstrapped confidence intervals over those by-participant means.

**Table 3. Helmert coding of reliability conditions.**

| Reliability | $C_1$ (reliable vs. unreliable) | $C_2$ (unreliable with bottom-up only vs. unreliable with instructions) |
|---|---|---|
| Reliable | 1 | 0 |
| Unreliable (with bottom-up only, Experiment 2) | -.5 | 1 |
| Unreliable (with instructions, Experiment 1) | -.5 | -1 |

impairments"). Participants' behavioral responses in Experiment 2 were strikingly similar to the unreliable speaker condition in Experiment 1, with anticipatory eye-movements within the pre-determined analysis window being significantly attenuated in comparison to the reliable speaker condition. To delve into possible differences between these three conditions, we conducted post-hoc analyses of offline (mouse-clicking) response times using the Helmert coding as described in Table 3. The results overall corroborated the fixation patterns (S2 Appendix). That is, response times were in general faster when the speaker was reliable ($\beta$ = -125.12, $t$ = -3.34, $p < 0.001$), and when instructions were heard in the 1-contrast condition as compared to the 2-contrast condition ($\beta$ = -19.59, $t = 2.41$, $p < 0.03$). There was a significant interaction such that participants more rapidly selected a referent in the 1-contrast condition when the speaker was reliable ($\beta$ = -125.12, $t$ = -3.34, $p < 0.001$). However, there was no significant difference between the two unreliable speaker conditions in Experiments 1 and 2.

These results together support the hypothesis that the exposure to pragmatically unreliable uses of language was sufficient to alter participants' real-time interpretations of scalar adjectives. Such results suggest that the modulation of contrastive inference, as examined here, is not strictly tied to the overt signaling of speaker unreliability, a feature typically unavailable in real-life scenarios. Rather, comprehenders seem to be sensitive to the linguistic evidence of pragmatic unreliability or uncooperativeness and are able to spontaneously alter their implicit interpretive behaviors in response to the observed idiosyncrasy. We must note, however, that the current statistical approach, based on the frequentist view on statistical significance, comes with clear limitations with respect to interpretations of null effects [52]. What we can report here is that the results from the current experiments provide reasonable evidentiary support for our main hypothesis, while replications of the results will be desirable.

One remaining question is about the speed at which the (un)reliability effect occurred in the present experiments. Did the participants in the unreliable speaker condition reduce their target fixations gradually as they observed over-informative expressions? Or were a few of such expressions at the outset of the experiment sufficient? Prior to the first critical item,

**Table 4. Summary of final linear mixed model of fixation proportions in Experiments 1 and 2.** Reliability conditions were Helmert coded (for details, see text).

| | $\beta$ | Std. Error | $t$- value | $p$-value |
|---|---|---|---|---|
| (Intercept) | -2.514 | .029 | -85.165 | <2e-16 |
| Reliability (reliable vs. unreliable) | .003 | .002 | 1.205 | n.s. |
| Reliability (unreliable bottom-up only vs. unreliable with instructions) | -.018 | .023 | -.80 | n.s. |
| Contrast (1-contrast vs. 2-contrast) | .012 | .014 | .881 | n.s. |
| Reliability (reliable vs. unreliable) * contrast (1-contrast vs. 2-contrast) | **.079** | **.017** | **4.623** | **< .0001** |
| Reliability (unreliable bottom-up only vs. unreliable with instructions) * contrast (1-contrast vs. 2-contrast) | .011 | .015 | .774 | n.s. |

Number of observations: 1152; Participants: 72; Items: 16.

participants observed four sentences within the video instruction, four example trials, and one or two filler item(s) depending on the condition (the number of filler items minimally varied according to list assignments, but they were all the kind where an over-informative adjective was inserted). Hence, participants in the unreliable speaker condition in Experiment 1 and those in Experiment 2 witnessed nine to ten utterances which included an over-informative scalar adjective by the time they encountered the first critical trial. Note that this is a post-hoc analysis, not planned prior to the data collection. Results of the analysis, therefore, should be interpreted as such and must be further investigated in a future study.

We used Generalized Additive Mixed Models (GAMM) [53], implemented with the *gamm4* package (ver. 0.2–5) conducted in R (S3 Appendix). In this model, we included the trial order as one of the predictors. In the reliable-speaker condition, proportions of target fixations within the expected time window (i.e., a 500ms window starting 200ms after the adjective onset) were higher in 1- than in 2-contrast trials. This effect was present from the earliest trial and gradually dissipated over the course of the experiment (similar effects of trial order were reported by the original study by Grodner and Sedivy (p.262-263) and by [19]). In the unreliable-speaker condition, however, such a difference between 1- and 2-contrast conditions was not observed even at the onset of the experiment. The presence and absence of the explicit instructions did not seem to significantly affect the bottom-up evidence required for modulation. Albeit preliminarily, the GAMM analyses thus support the idea that the nine to ten instances of over-informative usage of adjectives in filler trials at the onset of the experiment were sufficient to trigger the suspension of the contrastive inference.

## General discussion

Linguistic communication capitalizes on the impressive ability to map linguistic input onto the speaker's pragmatic intentions in context. Because speakers vary in their ways of encoding their intentions into words, it is considered effective for comprehenders to flexibly adjust their interpretation of this mapping according to *who* the input is coming from. Although talker-based adaptation has been attested in speech perception (e.g., [42, 43, 52–58]) and syntactic priming and processing (e.g., [59–62]), it has only begun to be understood with respect to pragmatics.

The current study examined whether, and if so how, real-time contrastive inference can be attenuated in response to the speaker's pragmatic unreliability. Extending Grodner and Sedivy [18], we manipulated the reliability in terms of the extent to which the speaker makes an informative use of scalar adjectives with respect to visual context. We found when the speaker was introduced as having linguistic impairments and failed to form informative referential expressions given a visual context, participants were less likely to generate contrastive inference based on a scalar adjective, as evidenced by early looks to the target referent in a size contrast pair. (Experiment 1). Crucially, we observed the same pattern without providing explicit information about the speaker, suggesting that bottom-up linguistic information was likely sufficient (Experiment 2). These findings lend support to the view that comprehenders can consistently and even spontaneously adjust their real-time pragmatic interpretation of utterances to accommodate the variability in language production across speakers [37].

Accommodating pragmatic idiosyncrasies of the speaker is often considered a process reserved typically for specific populations with underdeveloped linguistic commands (e.g., young children or non-native speakers) [63–68]. The current results, however, suggest that comprehenders were sensitive even to more subtle deviations from what is deemed pragmatically "optimal", such as using a linguistic form (e.g., the scalar adjective *large*) when it is, strictly speaking, redundant given a visual context. Detection of such deviations then led to

fine-grained adjustments of moment-by-moment interpretation of utterances. More specifically, comprehenders preferentially directed attention to a target referent in cases where the contrastive interpretation of an adjective would reliably lead to an intended referent (i.e., the reliable-speaker condition). In contrast, they delayed their commitment until the arrival of noun information when such anticipatory fixations could potentially result in a "detour" in directing attention to a target referent (i.e., the unreliable-speaker conditions). Although it has been suggested that implicit, error-based learning can explain such adaptive changes in other domains [59, 60, 69–71], much less is known about if the same mechanism could apply to pragmatic production and comprehension. Therefore, a major implication of the current results is that what appears to be instantaneous inferencing in reference resolution can reflect efficient use of cognitive resources given the current context. The current results thus open up an exciting avenue for future research about how contextual information can be tracked and used to modulate likely interpretations of intended meanings behind the chosen word in an utterance.

## How much bottom-up information is necessary?

Although the objective of the current study was to determine *whether* perceived pragmatic reliability would modulate contrastive inferences, it is important to ask about *the amount* of evidence necessary to elicit such modulation. How much evidence does it take the comprehender to modulate their processing behaviors? An answer to this question could distinguish contributions of two possible computational mechanisms potentially guiding the modulation. One possibility is that the speaker's pragmatic (un)reliability can be productively understood as distributional (statistical) learning based on observed linguistic forms [59, 69, 70]. Adult language users are known to be sensitive to statistics in the linguistic input, being able to match their internal predictions to the observed statistics e.g., [72]. It is possible that a similar mechanism allows comprehenders to track statistics of the speaker's referential expressions, including observed usage patterns of modifiers, conditioned on an identity of the speaker and a context [73–75]. Such learning has sometimes been stipulated to be "error based." That is, comprehenders may gradually alter their expectations for future input when their prediction turns out to be wrong, and this learning occurs in proportion to the magnitude of the errors experienced [59, 69, 70, 76]. In the current reliability manipulation, the instruction provided in the unreliable speaker condition in Experiment 1 could have facilitated such error-based learning, resulting in the modulation of the eye-movement patterns.

Another possible mechanism may involve a comprehender's causal reasoning about observed unreliability, beyond simply taking statistics (so called "data-explanation" approach) [77]. This view posits that comprehenders make sense of the input partly by "explaining" an unexpected instance of the input (e.g., Why did the speaker produce "large" in a given context?) and use the causal model to make predictions for subsequent input. An analogous case was made for listeners' strategies to cope with unreliable *pronunciations* of a speaker (e.g., *cigarette* as "/sh/igarette"): they can be attributed to a speaker-internal cause (e.g., language proficiency, regional accent, speech impediment), an incidental/situational cause (e.g., speech error, speaking with a pen in the mouth), or perception errors [78]. When the idiosyncrasy is likely attributable to the speaker, listeners would develop a speaker-dependent expectation for the subsequent input. Alternatively, when an unexpected input is likely caused by an incidental factor, it would not be expected to recur once the cause is removed [79, 80]. Contextually modulated interpretations of disfluencies (e.g., 'Click on [pause] thee uh. . .") have also been discussed under this model [25, 26, 81]. Unlike those of implicit statistical learning, effects of causal attribution come about instantaneously so long as the cause is unambiguous and readily accessible to the comprehender (e.g., perceiving a familiar foreign accent [46]).

The post-hoc analysis using a GAMM model provided an initial evidence of rapid attenuation of the contrastive inference. After listening to ~10 expressions that were not optimally economical, comprehenders did not seem to generate early fixations in response to a scalar adjective. This is more consonant with the causal attribution account rather than the implicit learning account, which draws on gradual accumulated knowledge of environmental statistics. As shown in past studies [22, 82–86], naturally produced referential expressions often contain more or less information than what is strictly necessary. If comprehenders are routinely attributing such variation to contextual causes, and if a speaker identity is regarded as a likely cause, a few over-informative instructions would be sufficient for them to make the link.

We emphasize again, however, that this question about the speed with which comprehenders determine (un)reliability of a speaker was not the focus of the current work. Additionally, recent results from a similar study suggest that the speed of pragmatic adaptation can depend on details of experiments (e.g., types of adjectives, visual stimuli) [19, 22]. In fact, the current data revealed that individual items and participants varied in their sensitivity to the reliability manipulation (Figs 1 and 2 in S2 Appendix). That is, some items (and or participants) were more responsible than others for the overall differences in the amount of anticipatory target fixations between the reliable vs. unreliable conditions. Note that these individual differences were appropriately taken into consideration as we entered them in all the mixed-effect models reported above. Nonetheless, further research is required to determine the amount and nature of the evidence that is necessary and how a potentially highly complex process of causal reasoning can be implemented in language processing.

## What is (un)reliability?

In the current experiments, the instructions by the unreliable speaker deviated from those by the reliable speaker by virtue of 1) over-informative instructions (i.e., 75% of their instructions had a superfluous modifier that is not strictly necessary to single out a target), 2) under-informative instructions (i.e., 12.5% lacked a modifier that would be necessary to pick out a target); and 3) erroneous instructions (e.g., 12.5% had a wrong noun such as "toothbrush" for "hairbrush"). We have implemented them to be faithful to the original study by Grodner and Sedivy [18], who adopted all three. The term *(un)reliability* was also introduced by them, and continued in the current study for the sake of consistency.

Evidently, these three types of referential expressions are pragmatically less than optimal for different reasons. Most importantly, over-informative instructions are perhaps non-economical/redundant/verbose but do not ultimately hinder the referential communication. The under-informative instructions, on the other hand, leave an intended referent unidentifiable and thereby lead comprehenders astray. The erroneous instructions would suggest that the speaker's difficulties may pertain not only to their use of context but also to their abilities to retrieve lexical information. Evidence of unreliability observed in the current experiment was thus not homogeneous in nature. Which of these instructions were critical to trigger the modulation in pragmatic processing?

Our decision to include a large number of over-informative instructions was based on their unique link to the early fixations that we examined here. We hypothesized that one type of information that comprehenders might assess across speakers and contexts is the *utility* of a given type of pragmatic inference. In the current scenario, contrastive inference in response to a scalar adjective (e.g., "Click on the large–") can accelerate the recognition of a target referent in the 1-contrast trial. This fast and accurate processing gives rise to the (expected) utility that can justify the cognitive, attentional, and other resources to be expended to derive inference. If so, among the three types of instructions produced by the unreliable speaker, over-informative

instructions clearly reduce the expected utility of contrastive inference, because now the inference does not result in benefits in processing. The under-informative and erroneous instructions, albeit problematic in their own ways, do not directly indicate this reduction of utility estimates in the given speaker's instructions. The absence of contrastive inference in Experiment 2 from an early point of the experiment suggests that the subtle indication of lowered utility of the contrastive inference would be enough to modify listeners' implicit, mostly automatic fixation behaviors.

If we instead provided more evidence of under-informative expressions and wrong nouns, listeners would have to lower the *overall* utility of the speaker's language use in general, beyond contrastive inference. That would, we predict, trigger different types of modulations of processing behaviors (e.g., disengagement, an overall slower reaction time). Future studies should manipulate the types and number of different signatures of "unreliability" to delve into the modulation of real-time pragmatic processing behaviors.

## Questions for future research

Here, we discuss two more open questions left for future research. The first question concerns whether, and if so to what extent, the current results truly reflect assumptions about the *speaker* and their overall pragmatic traits, as opposed to their uses of specific adjectives. We used only two types of scalar adjectives, both of which appeared in the critical as well as filler items. It is possible that participants assumed that the unreliability was confined to these two adjectives rather than a more general characteristic of the speaker affecting a wide range of lexical items. We believe that a line of ongoing work by Pogue and her colleagues is of relevance to this question.

Pogue et al. [28] conducted a series of offline judgment studies. Listeners were first exposed to two speakers who produced sentences with or without a scalar adjective, respectively, making the sentences either under-informative or optimally informative given a visual display (e.g., "Click on the {— / large} cup" when there was a large cup and a small cup). They were then presented with a written transcript of a sentence containing an under-informative color modifier (e.g., "Click on the {— / green} cup" when there was a large cup and a small cup, both green) and asked to select which speaker would be more likely to produce the sentence. They found that participants were more likely to choose the previously under-informative speaker, who did not use any modifiers, for the under-informative (i.e., color-modified) test sentences. Results such as these would suggest that listeners can and do track the likelihood at which the speaker provides a necessary and sufficient amount of information and generalize the assumption to unseen data. It is possible that a similar mechanism is at work during the online modulation of eye movements we observed in the current data.

Another question of importance is how exactly the pragmatic processing can be modulated based on the inferred reliability of the speaker [16]. Do comprehenders simply abandon ordinary comprehension behaviors altogether? Or is the modulation more targeted to linguistic elements (e.g., adjectives) with respect to which reliability was manipulated? The current study offers two insights that merit further explorations. First, participants' online eye-movements and offline picture-selection (mouse-clicking) response times (S4 Appendix) suggest that there was no significant delay in the responses in the unreliable-speaker conditions. As evidenced by the rapid increase in the target fixations in the region starting 200ms after the noun onset (Figs 4 and 5), participants in the unreliable-speaker conditions were quick to "catch up" by integrating the noun information as soon as it became available. While, as expected, response times were on average shorter in the reliable-speaker condition (Mean = 2,964ms) than in the two unreliable-speaker conditions (3,139ms in Experiment 1 and 3,106ms in Experiment 2),

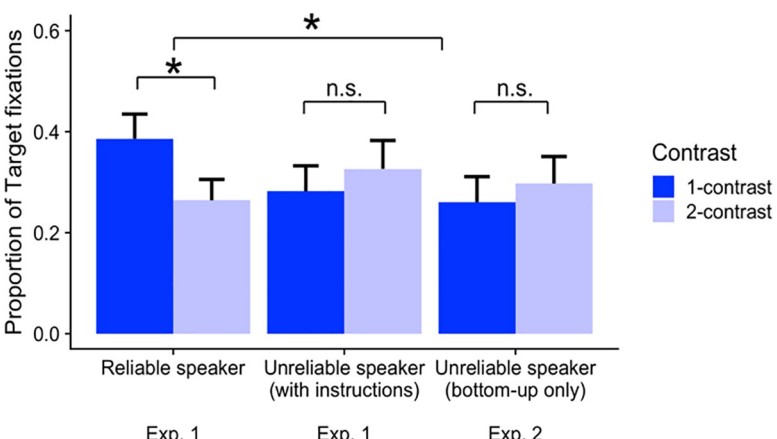

**Fig 5. Average proportions of target fixations during the 500 ms window starting 200 ms after the adjective onset (e.g., "Click on the large cup") by reliability and contrast conditions in Experiment 1.** Error bars represent 95% bootstrapped confidence intervals over those by-participant means.

the delay was deemed minimal (less than 200ms between the reliable- vs. unreliable-speaker conditions). There was virtually no clear sign of lapses of attention or disengagement consistent with the scenario where participants abandoned their ordinary, incremental processing behaviors.

However, an alternative strategy might have been devised in response to the unreliable speaker. We had expected that eye-movement patterns in the 2-contrast condition would be approximately equivocal in both reliable- and unreliable-speaker conditions. Instead, we found an interesting numerical trend in the two unreliable-speaker conditions such that participants were more likely to fixate on the target item over its competitor prior to the final noun (Figs 2 and 3). This observation is supported by the marginal simple effect of contrast in the unreliable-speaker found in Experiment 1. That is, the 2-contrast condition, compared to the 1-contrast condition, elicited a slightly higher degree of target fixations. This result is puzzling because there should be no linguistic information signaling an identity of the target in this time window. To confirm this, we conducted a follow-up survey with 42 participants who were not included in the eye-tracking experiments [48]. We presented a series of 4-image displays simulating our eye-tracking stimuli and played sound clips without the final noun (e.g., "Click on the large"). Participants selected one of the pictures based on these incomplete instructions. The mean proportion of selection of the target picture was 52%, suggesting that the linguistic information did not create a significant bias for the target item over the competitor item.

The result could potentially be due to co-articulation of an adjective-noun sequence or a subtle semantic congruency effect (e.g., *large* and *small* may be more likely to modify some nouns over others). In response to the pragmatically reliable speaker, such effects did not significantly affect the eye-movements: the target and competitor pictures were considered to be equally suitable candidates given the adjective information, as evidenced by the similar patterns of the target and the competitor fixations prior to the final noun (see top-right panel of Fig 2). In contrast, in the unreliable-speaker conditions, participants could have relied more on alternative sources of information to detect the target item at the earliest time point possible. Although the current study does not offer decisive insights into what these sources may be, they suggest the possibility that listeners begin to weigh other sources of information heavily when the speakers' ability to provide pragmatically well-formed instructions is called into question.

## Conclusion

The current study constitutes a first step towards understanding speaker-dependent modulations of real time, real world pragmatic processing. The results critically extend previous findings by illuminating a powerful inference architecture that may be at work to adapt moment-by-moment behavioral responses according to the reliability of linguistic cues estimated in the recent exposure. Future extensions of the current work will advance our knowledge about the computational and cognitive bases of pragmatic inferences guiding efficient and accurate language comprehension robust to variability in the input.

## Supporting information

**S1 Appendix. Explicit instructions in the reliable- and unreliable-speaker conditions.**
(DOCX)

**S2 Appendix. Data variations across participants and items.**
(DOCX)

**S3 Appendix. Post-hoc analyses of fixation proportions using Generalized Additive Mixed Effect Models (GAMMs).**
(DOCX)

**S4 Appendix. Mouse-clicking response times.**
(DOCX)

## Acknowledgments

Experimental design, subject testing, data analysis were conducted over the course of two years as part of the coursework of BCS 206–207 Undergraduate Research in Cognitive Science in the Department of Brain and Cognitive Sciences at the University of Rochester. We would like to thank the instructors of the course, Ralf Haefner and Florian Jaeger for their guidance and support, and the members of the course for their constructive comments when we were conducting this project. We would also like to acknowledge the audience of the 22nd annual AMLaP conference and the 92nd annual meeting of the Linguistics Society of America for helpful discussion points on an earlier version of this manuscript.

## Author Contributions

**Conceptualization:** Chigusa Kurumada.

**Data curation:** Sadie Dix.

**Formal analysis:** Chigusa Kurumada.

**Investigation:** Bethany Gardner, Rebecca Lawrence, Cameron Morgan, Anaclare Sullivan, Chigusa Kurumada.

**Project administration:** Bethany Gardner, Rebecca Lawrence, Cameron Morgan, Anaclare Sullivan.

**Resources:** Chigusa Kurumada.

**Supervision:** Chigusa Kurumada.

**Validation:** Chigusa Kurumada.

**Visualization:** Bethany Gardner, Chigusa Kurumada.

**Writing – original draft:** Bethany Gardner, Sadie Dix, Chigusa Kurumada.

**Writing – review & editing:** Chigusa Kurumada.

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
