## [Decision Letter · Decision Letter 0]

30 Jul 2020

PONE-D-20-20483

Online pragmatic interpretations of scalar adjectives are affected by perceived speaker reliability

PLOS ONE

Dear Dr. Kurumada,

Thank you for submitting your manuscript to PLOS ONE. After careful consideration, we feel that it has merit but does not fully meet PLOS ONE’s publication criteria as it currently stands. Therefore, we invite you to submit a revised version of the manuscript that addresses the points raised during the review process.

As you'll see, all three reviewers are very positive about your manuscript, but each offer suggestions for improvement. Reviewer 1 has questions and suggestions regarding your analyses as well as the manner in which your study is framed. Both reviewers 2 and 3 suggested moving some of the analyses you report in the General Discussion to the Results section (this is particularly relevant for the analysis of trial order given the importance of the bottom-up process you are examining). In addition, all reviewers provide multiple comments and questions for you to consider.  Finally, please add to your manuscript sentences describing where readers can access your data and materials (with links). For example, when discussing the analysis of Experiment 1, you could add a sentence saying, "The data used in this analysis are available at..." and then provide the link to that specific data file.

We look forward to receiving your revised manuscript.

Kind regards,

Thomas Holtgraves, Ph.D.

Academic Editor

PLOS ONE

Journal Requirements:

3. We note that Figure 1 in your submission may contain copyrighted images. All PLOS content is published under the Creative Commons Attribution License (CC BY 4.0), which means that the manuscript, images, and Supporting Information files will be freely available online, and any third party is permitted to access, download, copy, distribute, and use these materials in any way, even commercially, with proper attribution. For more information, see our copyright guidelines: http://journals.plos.org/plosone/s/licenses-and-copyright.

a)    You may seek permission from the original copyright holder of Figure 1 to publish the content specifically under the CC BY 4.0 license.

Reviewers' comments:

Reviewer's Responses to Questions

**Comments to the Author**

1. Is the manuscript technically sound, and do the data support the conclusions?

Reviewer #1: Yes

Reviewer #2: Yes

Reviewer #3: Yes

2. Has the statistical analysis been performed appropriately and rigorously? 

Reviewer #1: Yes

Reviewer #2: Yes

Reviewer #3: Yes

3. Have the authors made all data underlying the findings in their manuscript fully available?

Reviewer #1: Yes

Reviewer #2: Yes

Reviewer #3: Yes

4. Is the manuscript presented in an intelligible fashion and written in standard English?

Reviewer #1: Yes

Reviewer #2: Yes

Reviewer #3: Yes

5. Review Comments to the Author

Reviewer #1: This study shows evidence that listeners are sensitive to the extent to which speakers reliably use scalar adjectives as a disambiguating cue to reference. When a speaker says "point to the tall glass" in a context where there are two glasses (one tall and one short) and only says "point to the glass" in a context where there is only one glass, then listeners hearing "tall" are able to look at the appropriate glass even before they hear "glass". On the other hand, when a speaker says "point to the tall glass" all the time, listeners do not use "tall" as a cue to look at the tall glass right away. This study is essentially a conceptual replication, with improved methods, of previous work by Grodner & Sedivy; in addition to several purely methodological improvements, the authors also make a change to the instructions that allows them to show that listeners can figure out speaker reliability on their own without explicit instruction from the experimenters. Overall, the study is very well done and methodologically rigorous; I also greatly appreciate that the data, materials and analysis code have all been made available. I have just a few suggestions about improving the data analysis, and some conceptual comments/questions about the framing of the study. None of these are super major issues; I think this paper would be very appropriate for PLoS ONE.

-- Methods comments --

I'm confused about the authors' claims regarding the averages and confidence intervals calculated for Figure 2. The preceding text says "we calculated the proportion of fixations to the target, competitor, and contrast separately for each unique combination of participant and item" (a "unique combination of participant and item" is, in this context, just a trial, right?), then averaged across participants, and then made bootstrap CIs of those by-participant averages. The authors claim (lines 304-305) that this procedure ensures the CIs are not anticonservative. I'm not sure I follow that claim. First of all, when one has data with repeated measures for items as well as repeated measures for participants, then CIs calculated only on the basis of participants (on data aggregated across items) always have a risk of being conservative or anticonservative; in practice, there's usually more variation over participants than items, and thus by-participant CIs tend to be overconservative and by-item CIs tend to be anticonservative; but in theory, if a situation were to arise where there is more variation over items than over participants, then it's totally possible for by-participant CIs to be anticonservative; thus, I don't see how this procedure guarantees that CIs will not be anticonservative. Secondly, I'm not sure CIs of averages of this kind of data are very meaningful anyway, because I assume that the trial-level time bin averages are mostly 1s and 0s (the only time a given bin would have any other average within a single trial is if a saccade into or out of a region occurred during that very bin) and thus the data are not normally distributed anyway. Finally (and I think most importantly), this is kind of moot because I don't think CIs around individual conditions are of any interest here anyway; what one should really be interested in is the CI of the *difference* between conditions.

The use of an analysis window time-locked to adjective onset rather than offset looks like a definite improvement over the original G&S study. Nevertheless, this window still includes a lot of time after the noun has been heard (as shown in Figure 2). I guess the evidence for anticipatory fixations would be even stronger if it could be shown that these are happening before the noun is heard at all (or at least before 200 ms post-noun-onset, given the assumption that even after the noun onset it would still take 200 ms to program and execute a saccade to the noun). Do the results still hold up if you analyze this kind of shorter time window?

Is the recognition point of the adjective close to its onset? Is there ever any phonetic overlap between the adjective and the name of some competitors? For example, what if there's a display with a large cup (target), small cup (contrast), and lava lamp (competitor) -- I assume it would take some extra time to figure out if one is hearing "large" or "lava", and thus the onset of the adjective is not really the onset of when participants recognize they're hearing a scalar adjective here. Was anything done (either in stimulus design or in later analysis) to handle this kind of issue?

I think empirical logit is not recommended anymore (see e.g. https://www.sciencedirect.com/science/article/pii/S0749596X1630167X). What's really in vogue nowadays is growth curve modeling, although I'm not sure it's the best choice here (my understanding is that it's useful if you want to look at certain particular features such as comparing two conditions' slopes or curviness or asymptotes or whatever, which does not appear to be relevant for the hypotheses of interest here); for just overall checking if one condition is higher than another during the time window of interest, I think the best method is bootstrapped differences of time series (https://www.sciencedirect.com/science/article/pii/S0749596X18300470), which is basically just a cluster-based permutation test that's run after using a growth curve model to "smooth" the data. Alternatively, if more informative graphs of the data are included (graphs that show the by-participant and by-item variability; see https://www.polyu.edu.hk/cbs/sjpolit/pubs/447184_Politzer-Ahles_Manuscript.PDF#page=22 for what I have in mind, although this might be difficult to implement for the present study given that there are a lot of conditions that would need their own separate grahps) then readers would not need to get so hung up on stats since they would be able to see the pattern for themselves.

-- Moderate comments --

It would help if the authors could more explicitly foreground what's new about this study; in the current version, I didn't really figure it out (and I might still be wrong) until around pp. 6-7. They raise the question of whether comprehenders interpret stuff differently as a function of what they know about the speaker, but they also mention (e.g. line 122) several studies which seem to already show that this is the case. Is the important difference here the kind of pragmatic phenomenon being tested (e.g. that this study focuses on "the tall glass" sort of contrastive inference, whereas a lot of the previous ones focused on stuff like scalar implicatures about "some"), or the method, or something else? And why is this different thing worth focusing on (i.e. what does it tell us that the previous studies did not)? The clearest statement I see is the paragraph at the end of the introduction explaining that in previous studies the speaker reliability is explained through explicit instructions, whereas the present study will see if people figure out speaker reliability on their own. However, this reads as if it's sort of tacked on as an afterthought; the title and the abstract don't make it sound as if this is the main goal of the study (the abstract mentions the explicit instruction thing, but in passing, making it sound more like "here's a methodological problem with experiment 1 which we then fixed in experiment 2" rather than "this is the point of the whole study). (Also, if this is the main point of the study, my same question still comes up -- if I recall correctly, studies such as those by Fine & Jaeger have demonstrated that people can learn/adapt over the course of an experiment without explicit instructions [although I think recently there was a failure to replicate that somewhere? I can't recall the details], so why is it necessary to show it again in a new phenomenon / new manipulation?) If the stuff about the explicit instructions is the main motivation of the study, it would be useful to foreground this. If it's not (i.e. if something else is the main focus, and the explicit instructions stuff is a concern that came up during/after doing the first experiment), it would be useful to have a clear statement earlier of what the authors consider to be the main "new" thing addressed by this study. In any case, I'm sure the authors have a good reason for why it was important to test this particular phenomenon, I just struggled to identify it based on the manuscript as written. Ultimately, from the "Current study" section, my understanding is that this study is basically intended to be a replication of Grodner & Sedivy with improved methods, but I did not get that message from the Introduction. (I know at some journals there is an unreasonable pressure to present a study as being more novel than it is, but for PLoS ONE I think if the study really is an improved-replication of G&S then it would be ideal to say that outright; for PLoS ONE, novelty is not a major review criterion anyway, and more clearly foregrounding the motivation of the will make it easier to understand and evaluate the suitability of all the methodological details and the conclusions made from the results, and will keep the reader from mentally going off on tangents like I was along the lines of "didn't X et al already sort of test this?".)

According to the authors' description, the "unreliable" speaker was made unreliable mostly by virtue of giving overinformative instructions (75% of their instructions had a "superfluous" modifier, 12.5% were underinformative, and 12.5% had a wrong noun; if the wrong noun was always related to a single referent on the screen [e.g., "toothbrush" and "hairbrush" in the example given by the authors] then I guess the 4 underinformative trials are the only ones where the listener truly can't tell what to click). I'm not sure I would consider overinformative instructions evidence of an "unreliable" speaker. Overinformative instructions are not necessarily infelicitous; the "superfluous" modifier could be a non-restrictive adjective. For example, recently on Facebook I keep seeing ads along the lines of "You will LOVE this chonky seal", with a picture of a chonky seal toy; it's a no-contrast set (because there's not a skinny/normal seal next to the chonky seal), so I assume it's either meant as a non-restrictive modifier (like "this is a seal and it's also chonky") or it's meant to be interpreted in contrast with the set of other seals in the world not shown in the display (like "regular seals suck, but this one in the photo is chonky so you will like it"). The overinformative/superfluous stuff is the only stuff participants are guaranteed to see before any critical trials (since those were included in the introduction before the experiment, whereas underinformative or wrong-noun trials apparently were not; and assuming all trials were in a random order [although it sounds like they were in a controlled pseudorandom order, rather than fully random], the underinformative or wrong-noun trials may have come late in the experiment). Thus, the main thing we can assume the participants know about the speaker is that the speaker is not maximally "economical": they say words that might not be needed to uniquely identify the intended referent, but which might still have some other purpose (like the "chonky" in "chonky seal"). If all someone does is produce more adjectives than strictly necessary (again here I'm ignoring the underinformative and wrong-noun trials since they are rare and might not have been seen before the critical trials were seen) then they might be a perfectly reliable user of non-restrictive, flowery modifiers. Thus, I wonder if there's some other way this speaker could be described rather than "unreliable"; or if it could be revised to make sure that "unreliable" is referring not to the speaker itself but to the fact that the presence of an adjective in this speaker's speech is not a reliable signal that there is a contrast coming up.

The authors sometimes talk about listeners making anticipatory eye movements (e.g., lines 351-352: "participants were more likely to make anticipatory eye movements based on a scalar adjective in 1-contrast trials than in 2-contrast conditions"). However, the data analysis does not actually look at that, since what's analyzed here are fixations rather than saccades. One can conclude based on the analyses that participants were more likely to be looking at one thing than another, but not that they were move likely to "make anticipatory eye movements based on a scalar adjective" (that latter conclusion requires additional evidence, like observing that people were looking more at the target in this time window and they weren't in an earlier time window and thus the eyes must have moved somewhere in between; this stuff is visible in the figure but it's not actually addressed in the data analysis). There are analysis methods available to capture eye movements specifically (see e.g. Kingston et al. 2016: https://psycnet.apa.org/record/2016-39630-001) but I'm not sure that's really necessary here; an alternative solution would just be to rephrase all this to make sure it's referring to fixations rather than movements.

-- Minor comments --

line 116: Should this say "emerging" rather than "emergent"? To me, "emergent" means something is sort of epiphenomenal, naturally coming out of something else (i.e., it might already be done emerging; the point is about *how* it emerged), whereas "emerging" means something that is new, still in the process of emerging. In this context, I would interpret "an emergent body of work" as meaning something along the lines of "there were some studies testing other things, but an unexpected thing that comes out of them is this issue of whether comprehenders adapt...", which I don't think is what the authors meant.

line 122: Here the authors list a few studies showing that comprehenders' interpretations are modified by what they know/expect about the speaker. Another relevant study that should be mentioned here is Bergen & Grodner 2012. Similar to the Goodman & Stuhlmuller offline study and the Breheny & Ferguson eye-tracking studies, they found that comprehenders didn't seem to interpret "some" as "not all" when they were told that the speaker did not have full knowledge of what they're talking about.

--Stephen Politzer-Ahles

Reviewer #2: This struck me as generally a very sound piece of research and highly suitable for publication. There are some minor typos and trivial grammatical errors which could usefully be fixed, although they don't significantly detract from the readability of the work. My only substantive suggestion is that the analysis involving trial order, presently included as part of the General Discussion, could perhaps be placed with the rest of the statistical analysis of Experiment 2 - given that it's previewed by lines 381-2 I was expecting to find it occurring sooner.

Minor comments:

Lines 141-3: "It is therefore unclear under what circumstances, if ever, comprehenders conclude that they should withhold standard pragmatic assumptions and modulate their online linguistic comprehension accordingly." Could omit "if ever" - as I understand it, the preceding page discusses one such circumstance (i.e. when they are explicitly told that the speaker is likely to act in a non-standard way).

Lines 160-1: "This approach..." - I assume this is to be read as "The approach in the present paper" rather than referring to "the original study".

Lines 170-1: "neutralizing differences between the reliable and unreliable speaker conditions" - I assume that would be a conservative manipulation, in that it would tend to obscure differences that truly existed? It's perhaps worth being clear about that, as it sounds like a criticism of the original study on first reading.

Line 249: "the filler trials" - given that the treatment of these is part of the manipulation, should they be termed "filler"?

Line 253: "superfluous" - superfluous for the purpose of referring. I guess whether it's truly superfluous might depend on what instructions the speaker is presumed to be acting under.

Lines 346-8: This looks like a run-on sentence; is there a word missing?

Line 601: "for the under-informative (i.e., color-modified) test sentences." As I understand it, these sentences would be under-informative with or without color-modification: perhaps "incorrectly modified"?

Reviewer #3: The authors present a pair of studies investigating how perceived speaker reliability influences a comprehender’s pragmatic inferences. The first study is a conceptual replication of Grodner and Sedivy’s 2011 paper. The authors use a computer based eye-tracking study as opposed to the array manipulation design of G&S’s original study. There are two conditions: reliable and unreliable. In the unreliable condition participants are told that the study intends to “examine the communicative aspect of [the speaker’s] language impairment”. They find that when scalar adjectives are used by a perceived unreliable speaker this does not offer an advantage in referent identification, even in a condition where the adjective is sufficient to disambiguate the target item. With a reliable speaker this advantage is seen in an increase in anticipatory fixations to the target item. Thus replicating the findings from G&S but in a computer-based task. Experiment 2 builds on this by noting whether comprehenders need explicit instruction as to their interlocutor being an unreliable speaker or not. They use the same stimuli as in the unreliable condition of Experiment 1 but do not provide any indication as to the speaker’s pragmatic ability. The authors find no significant difference between the two unreliable conditions.

The paper itself is well written and presents two clear experiments that address an interesting question. It is clearly motivated and well situated within the literature. I appreciated the clarity of the methods and the analysis. The discussion of the results is good and I am impressed with the additional anaylses included and the clarity that they were post-hoc decisions. While the conclusion that comprehenders are sensitive to pragmatic unreliability/ uncooperativeness and are able to alter their behaviour accordingly is not surprising, it is nice to see a clear experimental demonstration supporting this.

Since the manipulation was between participants I do wonder whether participants had a strategy they used in the unreliable condition in that they knew the scalar adjective was unlikely to be a useful cue for disambiguation or whether their behaviour was specific to the unreliable speaker themselves. I don’t think it is possible to tease this apart with the data they have, but I would be intrigued to know if we are flexible in switching our processing depending on the person rather than the context. Although these are very similar things, I think my questioning is more about whether it is a more general processing shift that happens during the testing session vs a specific sensitivity to a certain person’s communicative style.

Minor points:

It might be worth noting the limitation in the statistical methods when obtaining null results; perhaps take a look at a Bayesian approach in the future?

In table 4 I think there’s an error in the second column. For the Reliable condition you’ve said the explicit instruction was “reliable (no mention of the communicative impairment)” and for Unreliable Exp2 you’ve said the explicit instruction was “reliable”. I think these should be the other way around?

I’m not entirely sure why the post-hoc analysis has been included in the general discussion rather than as its own section following on from Experiment 2- I think moving it may help with the flow of the paper.

Appendix 2 figure needs a legend.

6. PLOS authors have the option to publish the peer review history of their article (what does this mean?). If published, this will include your full peer review and any attached files.

Reviewer #1: **Yes: **Stephen Politzer-Ahles

Reviewer #2: **Yes: **Chris Cummins

Reviewer #3: **Yes: **Alice Rees

---

## [Author Response · Author response to Decision Letter 0]

11 Dec 2020

Thank you very much for your thoughtful reviews and suggestions. We have revised our manuscript thoroughly and addressed all the comments provided in the reviews. Our responses are detailed in the response letter. Thank you.

---

## [Editor Report · Decision Letter 1]

23 Dec 2020

Online pragmatic interpretations of scalar adjectives are affected by perceived speaker reliability

PONE-D-20-20483R1

Dear Dr. Kurumada,

We’re pleased to inform you that your manuscript has been judged scientifically suitable for publication and will be formally accepted for publication once it meets all outstanding technical requirements.

Kind regards,

Thomas Holtgraves, Ph.D.

Academic Editor

PLOS ONE
---

## [Editor Report · Acceptance letter]

2 Feb 2021

PONE-D-20-20483R1 

Online pragmatic interpretations of scalar adjectives are affected by perceived speaker reliability 

Dear Dr. Kurumada:

I'm pleased to inform you that your manuscript has been deemed suitable for publication in PLOS ONE. Congratulations! Your manuscript is now with our production department. 

Kind regards, 

on behalf of

Dr. Thomas Holtgraves 

Academic Editor

PLOS ONE